# Animating the Uncaptured: Humanoid Mesh Animation with Video Diffusion Models

**Marc Benedí San Millán, Angela Dai, Matthias Nießner**
Technical University of Munich, Germany
`{marc.benedi,`angela.dai,niessner`}@tum.de`

## Abstract

Animation of humanoid characters is essential in various graphics applications, but requires significant time and cost to create realistic animations. We propose an approach to synthesize 4D animated sequences of input static 3D humanoid meshes, leveraging strong generalized motion priors from generative video models – as such video models contain powerful motion information covering a wide variety of human motions. From an input static 3D humanoid mesh and a text prompt describing the desired animation, we synthesize a corresponding video conditioned on a rendered image of the 3D mesh. We then employ an underlying SMPL representation to animate the corresponding 3D mesh according to the video-generated motion, based on our motion optimization. This enables a cost-effective and accessible solution to enable the synthesis of diverse and realistic 4D animations. Project Website: https://marcb.pro/atu

## 1 Introduction

Character animation is fundamental in computer graphics – enabling lifelike, expressive, and engaging virtual characters for applications such as movies, video games, mixed reality, robotics, and many more. Such characters portrayed with realistic motions help to drive storytelling and interactivity, making crafted content more engaging and immersive.

Traditionally, character animation requires significant manual labor from highly-trained artists, who manually craft character rigs, define keyframes for motions, and fine-tune detailed motion behavior. This is both costly and requires a significant amount of tedious effort from skilled artists. Thus, leveraging learned motion priors to inform character animation would enable much more efficient animation synthesis. However, ground-truth capture of human motion is technically demanding and expensive to acquire, resulting in very limited data available (Mahmood et al., 2019; Liu et al., 2020; Zou et al., 2020) for training such motion priors, strongly limiting the diversity and generalization capability of methods fully supervised with such 4D data (Tevet et al., 2023; Shafir et al., 2024).

Recently, video diffusion models (Ho et al., 2022; Blattmann et al., 2023; Yang et al., 2024), trained on large-scale datasets with abundant video data, have demonstrated the ability to generate diverse and realistic videos conditioned on textual prompts. This suggests that these models implicitly learn motion priors that capture how the world evolves over time. Building on this insight, we propose a general approach for animating a 3D humanoid mesh based on a text description of the intended motion. Instead of relying on small and constrained 4D human motion capture datasets, we leverage motion priors learned by video generative models, which possess strong representational capacity and can synthesize diverse, high-fidelity human motion sequences.

Given a text prompt describing the intended motion, we generate a synthetic video of a 3D humanoid mesh performing the specified motion using a text-to-video (T2V) diffusion model. This model is conditioned on both the rendering of the 3D mesh and the text prompt. To facilitate motion tracking, we employ the SMPL body model (Loper et al., 2015) as a deformation proxy for the input mesh, enabling us to estimate the SMPL parameters and track the 3D mesh throughout the generated video frames.

Our approach begins by registering the SMPL body model to the input mesh, leveraging estimated body joint locations from multiple views. We then reparameterize the input mesh's vertex coor-

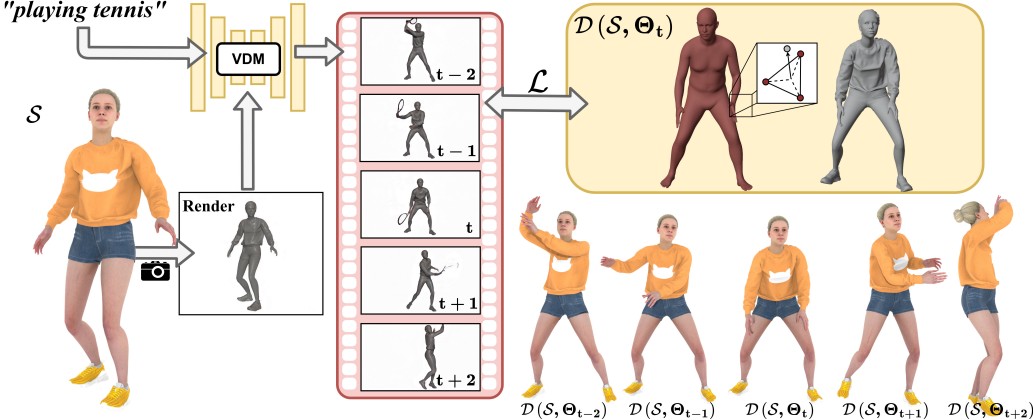

Figure 1: **Animating the Uncaptured**: A novel approach for text-driven animation of 3D humanoid meshes. Given an input mesh and a text prompt, we use a video diffusion model to generate a 2D video of the mesh performing the specified motion. This motion is subsequently extracted and transferred to the 3D geometry via sparse and dense tracking.

dinates into barycentric coordinates relative to the SMPL mesh faces. To transfer motion from the generated video to the input mesh, we extract 2D body landmarks, silhouettes, and dense DI-NOv2 (Oquab et al., 2023) features from the video frames, which serve as tracking cues for accurate motion reconstruction. This provides an easy, accessible approach to generate a wide range of realistic 4D humanoid animations.

Our contributions are summarized as follows:

- We tackle the task of mesh animation by leveraging the strong generative capabilities of text-to-video (T2V) diffusion models.
- We propose a pipeline to robustly track the motion from the generated video by combining body landmarks, silhouettes, and dense features.

## 2 RELATED WORK

**Text-to-Human Motion Generation** Human motion generation is a pivotal area in computer graphics, focusing on synthesizing realistic human movements for applications such as animation and virtual reality.

Although human motion can be captured using motion capture (MoCap) systems (O'Rourke & Badler, 1980; Ionescu et al., 2014; Von Marcard et al., 2018), this process requires specialized setups and actors which is expensive and time-consuming. This motivates the development of data-driven methods that can generate human motions from different input signals, such as actions (Guo et al., 2020), and audio (Fukayama & Goto, 2015; Crnkovic-Friis & Crnkovic-Friis, 2016; Deng et al., 2025) enabling the creation of diverse and realistic human animations without the need to manually capture every new motion.

In particular, text-to-human motion generation aims to synthesize human motion sequences based on text prompts describing the desired action in natural language. Zhang et al. (2024b) proposed the first text-conditioned diffusion model (Ho et al., 2020) for human motion generation. Such text-to-motion methods typically require paired text and body pose parameters for training, which are often obtained from motion capture data. However, such data is limited in quantity and diversity, and may not generalize well to unseen actions (Guo et al., 2022; Plappert et al., 2016; Guo et al., 2020).

Similar to our approach, MotionDreamer (Uzolas et al., 2025) proposes to extract motions from video diffusion models to animate meshes. As proposed by Tang et al. (2023); Zhang et al. (2023a); Luo et al. (2023), they extract semantic features from the intermediate activations of the diffusion model and perform feature matching between frames. This is an expensive process and limits their

pipeline to use low resolution features. In contrast, we employ both sparse and dense features for mesh registration and tracking, and our focus on humanoid meshes enables us to use stronger body priors for regularization. Similarly, AnyMoLe (Yun et al., 2025) leverages video diffusion models for motion in-betweening of arbitrary characters. Their approach involves fine-tuning the video model on rendered context frames and training a scene-specific joint estimator which is computationally expensive. In contrast, our method is designed for text-to-motion generation for humanoid meshes and does not require any fine-tuning of existing models.

**Human Pose and Shape Estimation from Images and Videos**   Monocular HPS estimation remains particularly challenging due to the inherent ambiguity in the 2D-to-3D mapping and the absence of depth information. To solve this, methods often rely on statistical body models (Loper et al., 2015; Osman et al., 2020; Xu et al., 2020; Pavlakos et al., 2019; Jiang et al., 2024) which provide a prior of shapes and poses and a low-dimensional parameterization of human body.

Two main approaches have been widely adopted to address this problem: optimization-based methods and regression-based methods. *Optimization-based* methods iteratively refine 3D pose and shape parameters by minimizing reprojection error between the 3D model and 2D image observations, such as silhouettes or body keypoints (Lugaresi et al., 2019; Cao et al., 2019; Pishchulin et al., 2016; Khirodkar et al., 2024). Bogo et al. (2016) proposes to predict 2D joints location from an image and then optimize the SMPL body prameters to minimize the reprojection error. To seek a stronger body prior, Pavlakos et al. (2019) trains a variational autoencoder (Kingma & Welling, 2013) to learn the representation of human poses. *Regression-based* methods leverage deep learning to directly estimate 3D pose and shape parameters from images or videos. Frameworks such as HMR (Kanazawa et al., 2018) and SPIN (Kolotouros et al., 2019) employ neural networks to predict SMPL parameters by learning from paired image data with pseudo-ground truth annotations. While these methods achieve real-time inference and improved robustness compared to optimization-based approaches, their performance is limited by the quality and diversity of the training datasets. As a result, regression-based models often struggle to generalize to out-of-distribution data, such as synthetic videos generated by text-to-video models.

Recent advancements have extended the image-based formulation into the video domain. By leveraging temporal information, these methods significantly improve the consistency and realism of estimated human motion across frames (Sun et al., 2024; Shin et al., 2024; Goel et al., 2023).

Such human pose estimation from images and videos have also been used to synthesize 3D and 4D human-object interactions (Li & Dai, 2024; Li et al., 2024). In this work, we use an optimization-based approach to focus on humanoid animation. Our approach robustly handles the synthetic videos generated by a text-to-video model to effectively generate diverse, realistic motion for various humanoid meshes.

## 3   METHOD

Our method tackles the task of text-to-motion for humanoid meshes. Given a text prompt $(\mathcal{P})$ describing the desired motion and an untextured humanoid mesh $(\mathcal{S})$ without any rigging or skeleton in an arbitrary pose, the method generates the deformation parameters $(\Theta)$ to animate the mesh over time. For this, we leverage the motion priors of Video Diffusion Models (VDMs) by generating a video conditioned on the prompt and the rendering of the mesh $\left(I_{t=0}^{\mathrm{RGB}} \in \mathbb{R}^{H \times W \times 3}\right)$. We use the SMPL body model Loper et al. (2015) as a deformation proxy to animate the input mesh through its parameters (section 3.3.1). We then optimize the deformation proxy parameters to match the motion in the generated video (section 3.3.2). The overview of our method is illustrated in figure 2.

### 3.1   PRELIMINARIES

**SMPL**   The Skinned Multi-Person Linear (SMPL) model (Loper et al., 2015) is a parametric body model that encodes body shape and pose variations using a learned low-dimensional representation. It is defined by the shape parameters $\beta \in \mathbb{R}^{10}$ and the pose parameters $\theta \in \mathbb{R}^{23 \times 3}$, where the shape coefficients correspond to the principal components of the body shape and the pose parameters encode the rotations of 23 skeletal joints in axis-angle representation.

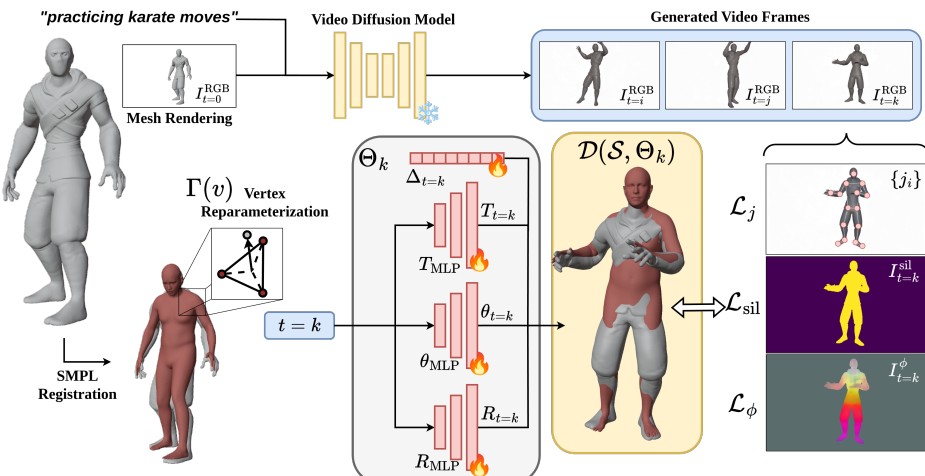

Figure 2: **Method overview.** Given an input mesh in an arbitrary pose and a text prompt describing the desired motion, we generate a video conditioned on the text prompt and the rendering of the mesh. We leverage the SMPL body model as a deformation proxy to track the motion from the video and transfer it to the input mesh. Specifically, we fit the SMPL model to the input mesh and associate its vertices with the SMPL faces (3.3.1). Finally, we optimize the SMPL parameters to match the video motion using estimated body landmarks, silhouettes and DINOv2 features extracted from the frames (3.3.2).

In this work, we use the SMPL model as a deformation proxy to animate the input mesh. To represent the pose parameters, we leverage the pre-trained variational autoencoder VPoser (Pavlakos et al., 2019), which provides a strong prior on valid human poses. Instead of directly optimizing the pose parameters $\theta \in \mathbb{R}^{23 \times 3}$, we optimize the latent encoding $Z \in \mathbb{R}^{32}$ of VPoser. This latent vector is then decoded to obtain the corresponding pose parameters. To simplify the notation, we use the symbol $\theta$ to refer both to the pose parameters and the VPoser encoding.

## 3.2 Video Generation and Frame Features

We start by normalizing $\mathcal{S}$ to unit scale and centering at the origin. We define $P : (\mathcal{S}, C) \to I \in \mathbb{R}^{H \times W \times 3}$ as the rendering function of the mesh $\mathcal{S}$ from camera $C$ to obtain the frame $I^{\mathrm{RGB}}$.

**Video Generation**  We use a Video Diffusion Model (VDM) to generate a video $\left\{ I_t^{\mathrm{RGB}} \right\}_{t=0}^{F-1}$ with $F$ frames depicting the mesh performing the motion described by the prompt. For this, we condition the VDM with the frame $I_0^{\mathrm{RGB}}$ and the prompt $\mathcal{P}$. Note that the first frame of the generated video is the same as the rendered image $I_0^{\mathrm{RGB}}$.

**Body Landmarks Estimation**  We use MediaPipe Pose Landmarker (Lugaresi et al., 2019) to estimate the body pose landmarks from the video. For each frame, MediaPipe provides 33 landmarks, $(j_i, \omega_i)$, where $j_i$ are the normalized pixel coordinates and $\omega_i$ are the confidence scores. We rearrange them to be in the same order as the SMPL joints, and apply a smoothing filter to mitigate the effect of noisy predictions.

**Dense Features**  Due to the sparse and noisy nature of landmark estimation, we additionally extract dense per-pixel features from the frames using a pre-trained DINOv2 model (Oquab et al., 2023) and obtain $\{I_t^{\phi}\}_{t=0}^{F-1}$. Following Dutt et al. (2024), we annotate the vertices of the mesh $\mathcal{V} = \{v_i\}$ with the features $\mathcal{V}^{\phi} = \{v_i^{\phi}\}$, where $v_i^{\phi}$ is the feature vector of vertex $v_i$. To compute these features, we render the mesh from 100 uniformly spaced cameras positioned on a sphere centered around the mesh, extract the features from each view, and back-project them for accumulation onto the mesh vertices. Finally, due to memory constraints, we apply PCA to reduce the dense features and keep the first 64 components.

**Silhouette** We extract the silhouette $I^{\text{sil}}$ of the mesh by thresholding the white background.

## 3.3 OPTIMIZATION

We reduce the task of transferring the motion from the video to the mesh as a video tracking problem. First, we register the SMPL (Loper et al., 2015) model to the input mesh and reparameterize $\mathcal{V}$'s coordinates with respect to the SMPL faces $\mathcal{F}^{\text{SMPL}}$ (3.3.1). Finally, we optimize the parameter $\Theta$ to deform the input mesh to match the motion in the video( 3.3.2).

**Model Parameters** The deformation function $\mathcal{D} : (\mathcal{S}, \Theta_t) \to \mathcal{S}_t$ is a function that deforms the input mesh $\mathcal{S}$ using the deformation parameters $\Theta_t$ for frame $t$. Note that for $\mathcal{D}(\mathcal{S}, \Theta_0) = \mathcal{S}$ due to the video generation process being conditioned on $I_0^{\text{RGB}}$. The parameters $\Theta$ are defined as $(s, \beta, \theta_t, T_t, R_t, \Delta_t)$, where $s$ is the scale of the SMPL model, $\beta$ is the shape parameters, and $\theta_t$, $T_t$, $R_t$, and $\Delta_t$ are the pose, translation, rotation parameters and per-vertex offsets for each frame $t$. Note that the shape and scale parameters are shared across all frames. For simplicity, we use $\Theta_t$ to refer to the deformation parameters for frame $t$: $\Theta = \{\Theta_t\}_{t=0}^{F-1}$.

To mitigate the influence of noisy signals during optimization, such as the jitter in the estimated landmarks, we opt to use shallow Multi-Layer Perceptrons (MLPs) to parameterize $\theta_t$, $T_t$, and $R_t$. That is, we use $\theta_{\text{MLP}}$, $T_{\text{MLP}}$, and $R_{\text{MLP}}$ to regress their corresponding SMPL parameters for each frame. As input, they take the Sinusoidal Positional Encoding (Vaswani et al., 2017) of the frame index. For simplicity, we use the same notation to refer to the SMPL parameters produced by the MLPs. For instance, $\theta_t = \theta_{\text{MLP}}(PE(t))$ represents the pose parameters predicted by the MLP for time step $t$.

### 3.3.1 SMPL REGISTRATION

We choose to leverage the SMPL model (Loper et al., 2015) for the following reasons: (1) due to monocular tracking being inherently ambiguous, we benefit from the body prior in SMPL and VPoser (Pavlakos et al., 2019), which ensures that the optimization remains within the plausible space of human poses; (2) the input mesh does not incorporate any deformation model such as skeleton or blend shapes.

We optimize for $s$, $\beta$, $\theta_0$, $T_0$, $R_0$; and minimize both terms in 1 and 2. The first term ensures that the SMPL joints $\hat{J}$ are close to the estimated 3D joint locations $J$ from the input mesh, while the second term minimizes the distance between the SMPL vertices and their nearest neighbors on the input mesh. To obtain $J$, we triangulate the 2D landmark predictions from MediaPipe (Lugaresi et al., 2019) using views sampled around the mesh.

$$\mathcal{L}_J = \frac{1}{N} \sum_{i=1}^{N} \omega_i \left\| \hat{J}_i - J_i \right\|_2^2 \tag{1}$$

$$\mathcal{L}_{\text{p2p}}(\mathcal{V}) = \frac{1}{|\mathcal{V}|} \sum_{v \in \mathcal{V}} \left\| v - \text{NN}(v, \mathcal{V}^{\text{SMPL}}) \right\|_2^2 \tag{2}$$

We also include squared $L_2$ priors for the shape and pose parameters, as defined in 3.

$$\mathcal{L}_\beta(\beta) = ||\beta||_2^2, \quad \mathcal{L}_\theta(\theta_t) = ||\theta_t||_2^2 \tag{3}$$

**SMPL as a Deformation Proxy** Finally, we reparametrize the input mesh vertices $\mathcal{V}$ with respect to the SMPL model. For this, after the registration process (3.3.1), we find for each vertex $v \in \mathcal{V}$ its closest SMPL face $f^{\text{SMPL}} \in \mathcal{F}^{\text{SMPL}}$. Then, each vertex $v$ can be represented using the barycentric coordinates of the corresponding SMPL face, along with the distance to the face plane:

$$v = \gamma_1 v_1^{\text{SMPL}} + \gamma_2 v_2^{\text{SMPL}} + \gamma_3 v_3^{\text{SMPL}} + d\mathbf{n} \tag{4}$$

where $\gamma_i$ are the barycentric coordinates of the input mesh vertex $v$ with respect to the SMPL face $f^{\text{SMPL}}$ formed by the vertices $v_1^{\text{SMPL}}$, $v_2^{\text{SMPL}}$, and $v_3^{\text{SMPL}}$, $d$ is the distance from the input mesh vertex

to the SMPL face, and $\mathbf{n}$ is the normal of the SMPL face. We refer to this differentiable function, which computes the location of the input mesh vertices from the SMPL vertices as $\Gamma : \mathcal{V}^{\text{SMPL}} \to \mathcal{V}$.

This ensures that the input mesh vertices are attached to the body model, allowing us to optimize the model parameters to match the motion in the video. To address potential outliers, we apply a robust filtering mechanism that identifies and excludes mismatches. Outliers are detected by combining multiple criteria: (1) absolute distance between the input mesh vertices and the SMPL model; (2) the angular deviation between vertex normals and face normals of the closest SMPL triangle, ensuring alignment within a threshold of 45 degrees; (3) neighborhood statistics, which analyze the mean and standard deviation of distances between vertices to identify points that deviate significantly from their neighbors.

### 3.3.2 VIDEO TRACKING AND MOTION TRANSFER

We transfer the motion from the generated video to the input mesh by optimizing the following parameters of the deformation model $\Theta_t$: $T_t$, $R_t$, $\theta_t$, and $\Delta_t$, keeping fixed the shape and scale parameters $\beta$ and $s$. We formalize the deformation model as follows:

$$\mathcal{D}\left(\mathcal{S}, \Theta_t\right) = \Gamma\left(s \cdot R_t \cdot \text{SMPL}\left(\beta, \theta_t\right) + T_t\right) + \Delta_t \tag{5}$$

**Loss terms** We optimize the parameters described above to minimize equation 6.

$$\mathcal{L}_{\text{total}} = \frac{1}{F} \sum_{t=1}^{F-1} \left(\mathcal{L}_j + \mathcal{L}_{\text{sil}} + \mathcal{L}_\phi\right) + \mathcal{L}_{\text{regs}} \tag{6}$$

The first data term, $\mathcal{L}_j$, shown in equation 7, minimizes the distance between the re-projected SMPL joints ($\hat{j}$), and the predicted 2D landmarks ($j$), where $w$ is the confidence score of the predicted landmarks, $\rho$ is the German-McClure loss function (Geman, 1987), and $N$ is the number of landmarks.

$$\mathcal{L}_j = \frac{1}{N} \sum_{i=1}^{N} w_i \rho\left(\hat{j}_i - j_i\right) \tag{7}$$

The second data term, $\mathcal{L}_{\text{sil}}$, shown in equation 8, minimizes the binary cross-entropy loss between the rendered silhouette ($\hat{I}^{\text{sil}}$) and the silhouette extracted from the generated video ($I^{\text{sil}}$), where $N = HW$ is the number of pixels.

$$\mathcal{L}_{\text{sil}} = -\frac{1}{N} \sum_{i=1}^{N} \left(I_{t,i}^{\text{sil}} \log(\hat{I_{t,i}^{\text{sil}}}) + (1 - I_{t,i}^{\text{sil}}) \log(1 - \hat{I_{t,i}^{\text{sil}}})\right), \tag{8}$$

The final data term, $\mathcal{L}_\phi$, shown in equation 9, minimizes the cosine similarity between the rendered features ($\hat{I}^\phi$) and the dense features extracted from the generated video ($I^\phi$).

$$\mathcal{L}_\phi = \frac{1}{N} \sum_{i=1}^{N} \left(1 - \frac{\hat{I}^\phi_{t,i} \cdot I^\phi_{t,i}}{\|\hat{I}^\phi_{t,i}\|_2 \|I^\phi_{t,i}\|_2}\right), \tag{9}$$

The last term, $\mathcal{L}_{\text{regs}}$, includes all regularization terms: $\mathcal{L}_{\text{temp}}$, $\mathcal{L}_{\text{ex. ben.}}$, and $\mathcal{L}_\theta$, which are defined in equations 10, 11, and 3, respectively.

Temporal regularizers are used to ensure smooth motion across frames and mitigate the impact of landmark jitter. In particular, we penalize abrupt changes in translation, rotation, pose parameters, and 3D joint locations between consecutive frames. The temporal regularization terms are defined as follows:

$$\mathcal{L}_{\text{temp}}(x) = \sum_{t=1}^{T} \|x_t - x_{t-1}\|_2 \tag{10}$$

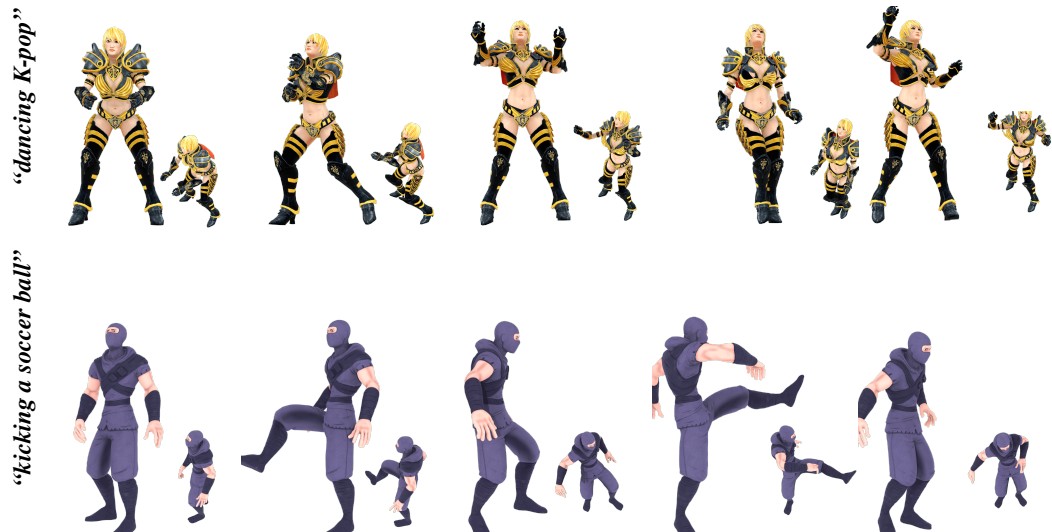

Figure 3: **Qualitative results**. Visualization of generated mesh animations with our method. Each row shows: the text prompt, the input mesh, and intermediate frames of the generated motion. For all generations, we visualize the frames from the front and side views.

Inspired by Bogo et al. (2016), we also include a term to penalize extreme bending of the knees and elbows. See equation 11. This term is defined as the sum of the squared angles between the upper and lower limbs, ensuring that implausible poses with excessive bending are avoided.

$$\mathcal{L}_{\text{ex. ben.}}(\theta) = \sum_{i \in (\text{elbows, knees})} \exp\{\theta_i\} \tag{11}$$

Additionally, as described in equation 3.3.1, we penalize deviations from the manifold of valid human poses by using the VPoser regularization term defined in equation 3.

Finally, we employ As-Rigid-as-Possible (Sorkine & Alexa, 2007) regularization to ensure that the resulting mesh deformation is smooth and preserves the mesh's intrinsic structure.

**Feature Mapper** Because the appearances of the mesh $\mathcal{S}$ and the generated video $\left\{I_t^{\text{RGB}}\right\}_{t=0}^{F-1}$ diverge over time, we optimize a learnable projection that maps the vertex features $\mathcal{V}^\phi$ into a space better aligned with the features extracted from the video $I^\phi$. This projection is optimized in a self-supervised manner.

## 4 EXPERIMENTS

### 4.1 IMPLEMENTATION DETAILS

In our implementation, we use PyTorch 2.0.1 (Paszke et al., 2017) and CUDA 11.7 (Nickolls et al., 2008), and perform differentiable rendering with PyTorch3D 0.7.7 (Ravi et al., 2020). We use SMPL (Loper et al., 2015) and VPoser (Pavlakos et al., 2019) as body priors. For frame processing, we use MediaPipe (Lugaresi et al., 2019) and DINOv2 (Oquab et al., 2023) to extract landmarks and dense features respectively. Our neural parameterization consists of shallow multilayer perceptrons (MLPs) with four layers and 128 hidden units per layer. We use a 64-dimensional positional encoding of the frame index as input. We used Kling AI (Platform, 2024) as a Video Diffusion Model. It generates 5-second videos at a resolution of $768 \times 1280$ pixels, which we downsample to $384 \times 640$ pixels during optimization due to memory constraints. For SMPL registration, as described in section 3.3.1, we set the batch size to 8 and optimize for $1,000$ iterations or until convergence, using a learning rate of $0.01$ with the Adam optimizer (Adam et al., 2014). Similarly, for video tracking, as described in section 3.3.2, we use the same batch size and optimize for $4000$ iterations or until convergence, with a learning rate of $0.001$. The complete optimization process takes approximately

Table 1: Pose fitting performance comparison with SMPLIFY-X (Pavlakos et al., 2019), it's smoothed version SMPLIFY-X*, WHAM (Shin et al., 2024), Multi-HMR (Baradel* et al., 2024),and our proposed method on untextured sequences from the CAPE dataset. Metrics include Mean-Per-Joint-Position-Error (MPJPE), Per-Vertex-Error (PVE), and acceleration error (Accel). Lower values indicate better performance across all metrics.

| Method | MPJE | PVE | Accel |
|---|---|---|---|
| SMPLIFY-X | 0.054 | 0.057 | 20.57 |
| SMPLIFY-X* | 0.053 | 0.056 | 02.18 |
| WHAM | 0.051 | 0.054 | 08.61 |
| Multi-HMR | 0.045 | 0.510 | 09.03 |
| Ours | **0.036** | **0.041** | **01.49** |

1.5 hours on an NVIDIA RTX 2080 Ti GPU (12 GB VRAM), 16GB RAM, and 4 CPU cores on an Intel(R) Xeon(R) Gold 6240.

## 4.2 VIDEO TRACKING

We evaluate the tracking performance of our method using rendered videos of animated untextured meshes, as this enables evaluation with ground truth while also mimicking the output of a text-to-video model conditioned on untextured meshes.

**Datasets** For evaluation, we use the CAPE dataset (Ma et al., 2020; Pons-Moll et al., 2017), which provides 4D sequences of clothed humans in motion. Each sequence includes a 3D human body mesh with the corresponding SMPL+D registration. We use these SMPL parameters as ground truth for evaluation. We evaluate a random set of 26 sequences from the CAPE dataset, totaling 7,000 frames, and render their untextured SMPL meshes.

**Baselines** We compare our method with two learning-based methods, WHAM (Shin et al., 2024) and Multi-HMR (Baradel* et al., 2024), and an optimization-based method, SMPLIFY-X (Pavlakos et al., 2019). Since SMPLIFY-X does not enforce temporal regularizations, we apply a smoothing filter to its results to improve the performance on video sequences. Finally, as our method uses the input mesh to get an initial alignment for the first frame of the sequence, we align the results of WHAM and SMPLIFY-X to the input mesh using Procrustes alignment as well for fair comparison.

**Evaluation Metrics** We evaluate the methods using three metrics: Mean-Per-Joint-Position-Error (MPJPE), Per-Vertex-Error (PVE), and acceleration error, which quantifies the inter-frame smoothness of the reconstruction. We follow established evaluation from Pavlakos et al. (2019), and include acceleration error as in Shin et al. (2024) to measure the temporal consistency.

**Results** As shown in table 1, our method outperforms SMPLIFY-X, SMPLIFY-X with smoothing, Multi-HMR and WHAM in all metrics. We note that Multi-HMR and WHAM, as learning-based methods, are not trained for tracking untextured videos, which causes its performance to degrade from its original setting. Our approach is better suited for tracking videos generated by a text-to-video model conditioned on untextured meshes.

## 4.3 MOTION FROM GENERATED VIDEOS

**Perceptual Study** We evaluate the quality of the motions generated by our method by conducting a perceptual study. A total of 30 participants took part in the study and were asked to answer the following questions: (Q1:) "Which motion best aligns with the given text prompt?", (Q2) "Which method generates more natural motions?", (Q3) "Which motion do you prefer overall?". Additionally, we evaluate the perceived quality of the motions using a Likert scale asking participants to rate the motions realism and prompt alignment from 0 (poor) to 5 (excellent).

For each test case, we generated a motion using our method, MDM (Tevet et al., 2023), FineMoGen (Zhang et al., 2023c), MotionDiffuse (Zhang et al., 2024b), ReMoDiffuse (Zhang et al.,

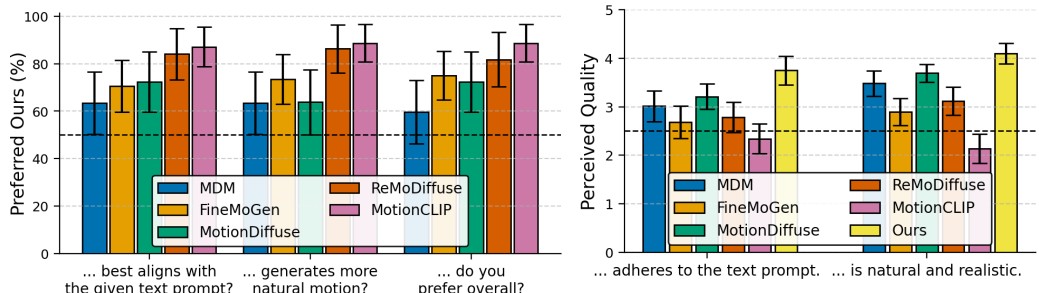

Figure 4: **User study results**. (Left) User study indicating the percentage of users that prefer our method over baselines. (Right) Perceived quality of the generated motions, where 5/0 indicate strong agreement/disagreement with the statement: *"The video..."*

*"bouncing a ball"*

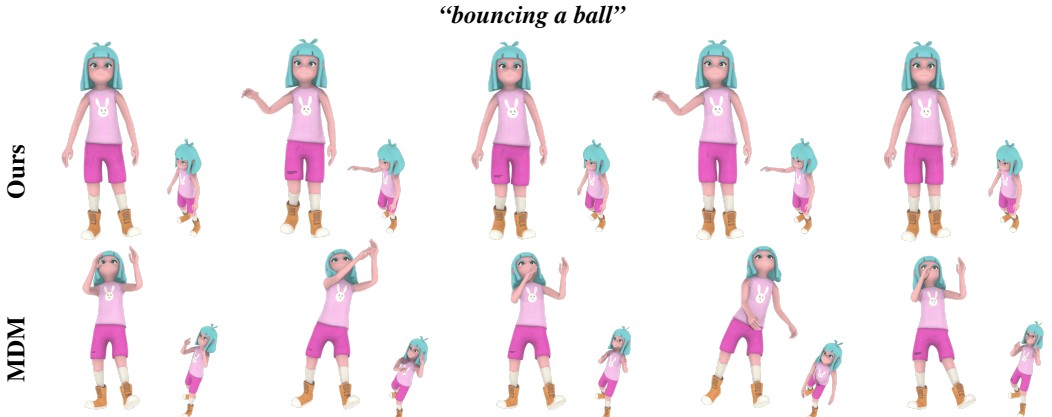

Figure 5: **Qualitative evaluation**. We compare the motions generated by MDM (Tevet et al., 2023) and our method for some of the prompts used in the perceptual study. We show two views (front and side) of the generated motions for multiple frames.

2023b), and MotionCLIP (Tevet et al., 2022) . Participants were shown 17 such motion pairs, each presented from two different views: front and side. To mitigate bias, the order of the motions was randomized for each participant. Note that the questions aim to not only evaluate prompt alignment but also the overall quality of the generated motion.

Figure 3 shows that participants prefer our method over the other baselines across all questions. We also observe that our method achieves higher scores in the Likert scale for both realism and prompt alignment.

**Qualitative Results**    In figure 5, we show a comparison of the generated motion using our method and MDM (Tevet et al., 2023) for the same motion prompts. We observe that our method generates motions that align better with the text prompt and look more realistic. Specifically, by leveraging the wide prior of the VDM, we can generate more diverse and realistic motions.

In figure 3, we show additional qualitative results of our method. We generate motions given a text prompt and visualize the frames from the front and side views.

## 4.4    LIMITATIONS AND FUTURE WORK

While video diffusion models (VDMs) show great promise in generating diverse and realistic human motions, they remain susceptible to artifacts such as morphing effects. However, with the rapid advancement of this field, we anticipate that future iterations of these models will address these shortcomings and further enhance motion realism. Monocular tracking remains an inherently underconstrained problem, often leading to ambiguities and inaccuracies in the reconstructed motion.

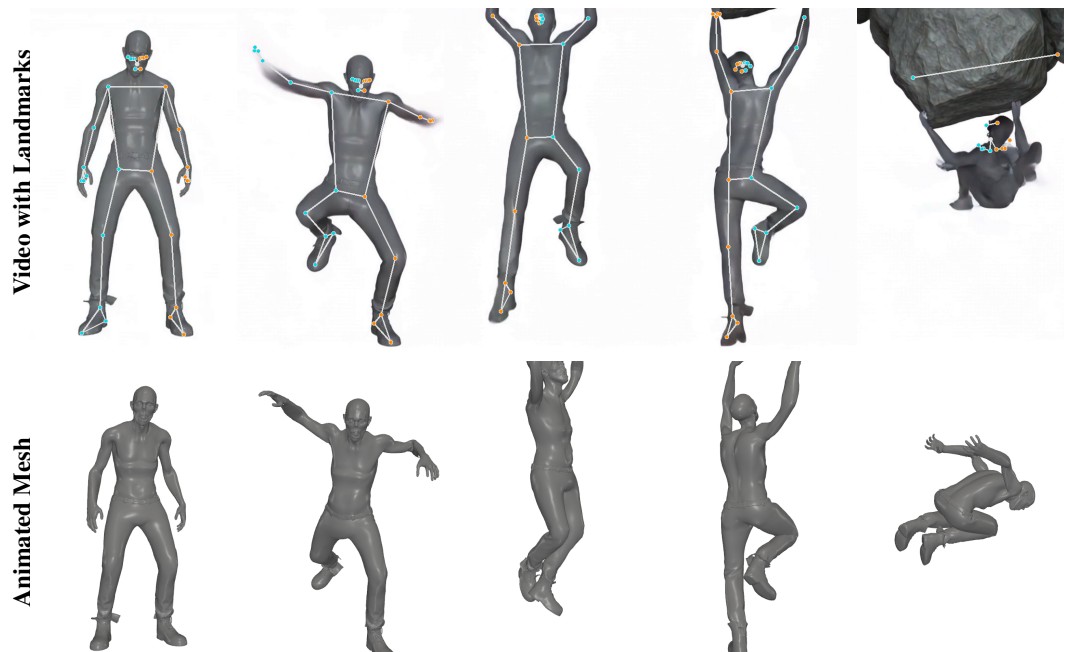

Figure 6: **Failure case.** Example of a failure case for the action "bouldering" where the generated video suddenly morphs the front of the body into the back. Our method solves this by smoothly rotating the mesh to accommodate the sudden change. At the end, the generated video fails to represent the mesh accurately which can be seen in the last column. However, this still does not collapse the final mesh.

To mitigate these challenges, future work could explore the integration of depth predictors (Khirodkar et al., 2024) or leverage multi-view diffusion models (Xie et al., 2024; Zhang et al., 2024a; Wu et al., 2025). Our proposed approach opens up exciting possibilities for generating 4D datasets of human motion. These datasets could serve as valuable resources for training and benchmarking models in human motion analysis. Moreover, our method has significant potential for practical applications, such as creating animations for virtual characters in video games, movies, and mixed-reality experiences.

#### 4.4.1 FAILURE CASES

As discussed in section 4.4, our method is limited by the capabilities of the underlying video diffusion model. This includes generating animations that adhere closely to the provided prompt when the input video does not accurately reflect the desired action. This can lead to tracking failures, especially in scenarios where the video contains morphing effects. We show an example in figure 6, where the sudden morphing of the body from front to back becomes a 180 degrees rotation in the generated animation.

## 5 CONCLUSION

In this work, we propose *Animating the Uncaptured*, a novel text-to-motion generation method for humanoid meshes. By leveraging the strong priors of video diffusion models, our approach generates realistic and diverse human motions, which are then transferred to 3D meshes. We use the SMPL model as a deformation proxy, anchoring the vertices of the input mesh to their closest SMPL face and optimizing the SMPL parameters to track the motion depicted in the generated video. This process is guided by extracting 2D body landmarks, silhouette information, and dense semantic features from the video frames. Experiments on the CAPE dataset demonstrate that our method quantitatively outperforms baseline approaches, particularly in tracking videos with untextured meshes. Finally, our user study highlights a strong preference for the motions produced by our method, both in terms of realism and alignment with text descriptions.

ACKNOWLEDGEMENTS

This work was supported by the ERC Consolidator Grant Gen3D (101171131) of Matthias Nießner and the ERC Starting Grant SpatialSem (101076253) of Angela Dai.

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

## A  SUPPLEMENTARY MATERIAL

We provide additional details about our method and results in this appendix. Additionally, we include a video in the supplementary material and an anonymous link to the source code: `https://anonymous.4open.science/r/aj93jkfjl8/`.

### A.1  ABLATIONS

Table 2: Ablation study on the effect of various components on performance. Metrics include MPJPE (Mean Per Joint Position Error), PVE (Per Vertex Error), and Accel (Acceleration Error). Lower is better for all metrics.

| Method | MPJPE | PVE | Accel |
|---|---|---|---|
| w/o $\mathcal{L}_\theta$ | 0.0486 | 0.0556 | 1.7121 |
| w/o $\mathcal{L}_{\text{ex. ben.}}$ | 0.0393 | 0.0453 | 1.5699 |
| w/o $\mathcal{L}_\phi$ | 0.0392 | 0.0447 | 1.5895 |
| w/o $\mathcal{L}_{\text{temp}}$ | 0.0403 | 0.0458 | 3.2005 |
| Opt. Parameters | 0.0453 | 0.0533 | 2.6494 |
| **Ours** | **0.0362** | **0.0411** | **1.4981** |

We conduct different ablation studies to evaluate the impact of our various design choices. To evaluate a setting where ground truth is available, we run the same evaluation as for tracking ( section 4.2) and show ablation results in table 2. We observe that the temporal loss $\mathcal{L}_{\text{temp}}$ clearly improves performance, particularly in acceleration error. *Opt. Parameters* refers to directly optimizing the SMPL parameters instead of using an MLP to predict them. This confirms that by using a neural parameterization, we can take advantage of the inductive bias of the MLP to improve the tracking performance, particularly, the smoothness. $\mathcal{L}_{\text{ex. ben.}}$ and $\mathcal{L}_\theta$ allow us to leverage the body prior, which helps to maintain anatomically plausible poses.

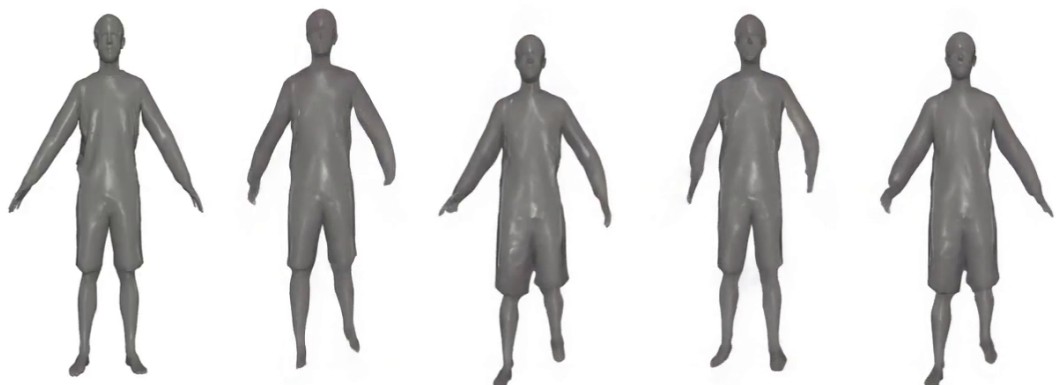

Figure 7: Example of a video generated with CogVideoX (Yang et al., 2024) conditioned on the rendering of a mesh and the prompt "The person is running".

## A.2 FULL PROMPT

We provide the prompt used to generate our results, where we replace the *ACTION* placeholder with the action we want to generate. The prompt emphasizes dynamic motions, human body realism, static camera and bright lighting. These are to avoid static videos, body morphing, camera zooms and dark lighting respectively. The prompt is as follows:

*"An award winning documentary about a person ACTION. Energetically ACTION. Dynamic movement. Light grey person. Realistic movement. Realistic motion. Realistic human body. Wide angle shot showcasing the man, dynamic movement, this video is incredibly detailed and high resolution, the uniform light is impressive, a masterpiece. Clear illumination. Bright light. No dark light. Fixed camera. No zoom in."*

## A.3 PERCEPTUAL STUDY

A total of 30 users participated in the study and gave consent for their data to be used for research purposes. We include a screenshot of the user study interface in figure 9.

### A.3.1 LIST OF PROMPTS

We list the prompts used in the perceptual study:

- *"The person is taking cover and shooting"*
- *"The person is practicing with nunchaku"*
- *"The person is playing tennis"*
- *"The person is practicing karate moves"*
- *"The person is taking cover"*
- *"The person is adjusting her glasses"*
- *"The person is stopping the traffic"*
- *"The person is digging a hole with a shovel"*
- *"The person is dancing disco fox"*
- *"The person is opening a door"*
- *"The person is tying her shoes"*
- *"The person is brushing her hair"*
- *"The person is mining with a pickaxe"*
- *"The person is smoking"*

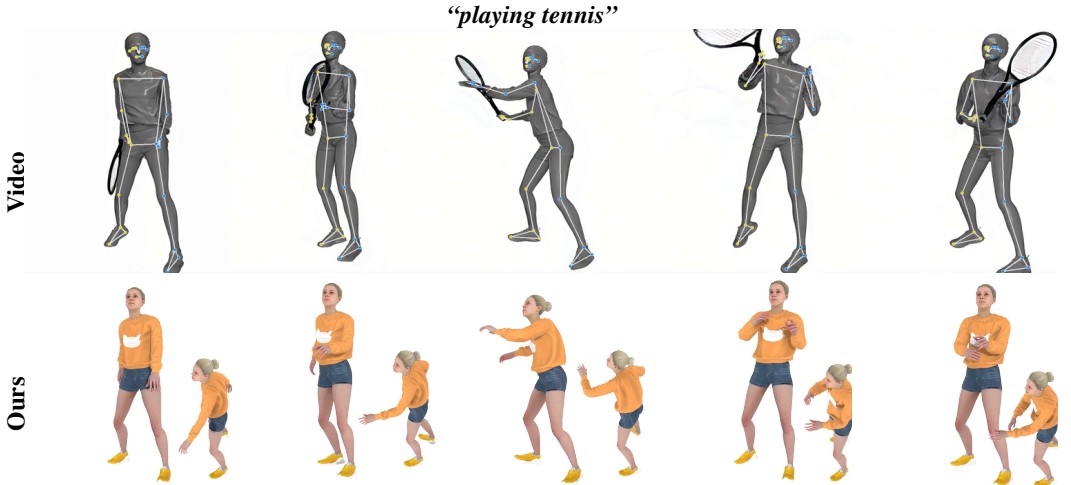

Figure 8: Example of a video generated with Wan2.2 (Wan et al., 2025) and the tracked mesh.

- *"The person is getting ready to fight"*
- *"The person is bouncing a ball"*
- *"The person is practicing kickboxing"*

### A.4 JUSTIFICATION OF THE CHOICE OF VDMS

The results of our method are generated using Kling AI (Platform, 2024), a closed-source model. While we believe that academic research should publish code and models to ensure reproducibility, we chose to use this model due to its superior performance on our task. However, we want to emphasize that our method is agnostic to the choice of VDM and can be used with any model that generates videos from text prompts, therefore, with the fast pace of research in this area, we expect that our method can be easily adapted to future models such as VideoJAM (Chefer et al., 2025). We show an example of the generated video obtained from CogVideoX (Yang et al., 2024) in figure 7.

#### A.4.1 WAN2.2

With the recent release of Wan2.2 (Wan et al., 2025), we were able to test our method with this open-source model. We provide an example of the generated video and tracked mesh in figure 8. We used a similar environment as in section 4.1, with the exception of using an NVIDIA A100 GPU with 80GB of memory.

### A.5 ETHICAL CONSIDERATIONS

We present a method for generating mesh animations from text prompts by leveraging video diffusion models. As a result, our approach inherits the ethical considerations associated with these models, particularly the potential for generating deepfakes of people performing actions. However, since our method requires a 3D mesh as input— which is costly and challenging to obtain for arbitrary individuals— we do not anticipate its use for creating deepfakes. Nevertheless, we acknowledge the importance of considering potential misuse and emphasize the need to raise awareness of the broader ethical implications.

### A.6 ADDITIONAL RESULTS

#### A.6.1 VIDEO TRACKING

We provide additional results of the video tracking on the Mixamo (Mixamo, 2025) dataset, which contains a variety of characters and motions for rigged humanoid meshes. For this, we used 50

Table 3: Per-Vertex-Error (PVE) on riggable untextured sequences from Mixamo (Mixamo, 2025) dataset for WHAM (Shin et al., 2024), SMPLIFY-X (Pavlakos et al., 2019) and our method.

| Method | PVE |
|---|---|
| SMPLIFY-X | 0.099 |
| WHAM | 0.066 |
| Ours | **0.058** |

randomly selected animations from the dataset, rendered them as videos, and tracked them using our pipeline. We report the average per-vertex-error (PVE) in table 3. The results demonstrate the effectiveness of our video tracking method on diverse characters and motions.

### A.6.2 QUALITATIVE RESULTS

We provide additional results of our method in figures 10 and 11, and additional comparisons against MDM in figure 12.

Figure 9: **Perceptual Study Visualization**. A screenshot of the user study interface. The user is presented with a text prompt and two methods to compare. Each method has two videos, one from the front and one from the side. The user can play the videos and compare them side by side. At the bottom of the page, the user has to answer a set of questions.

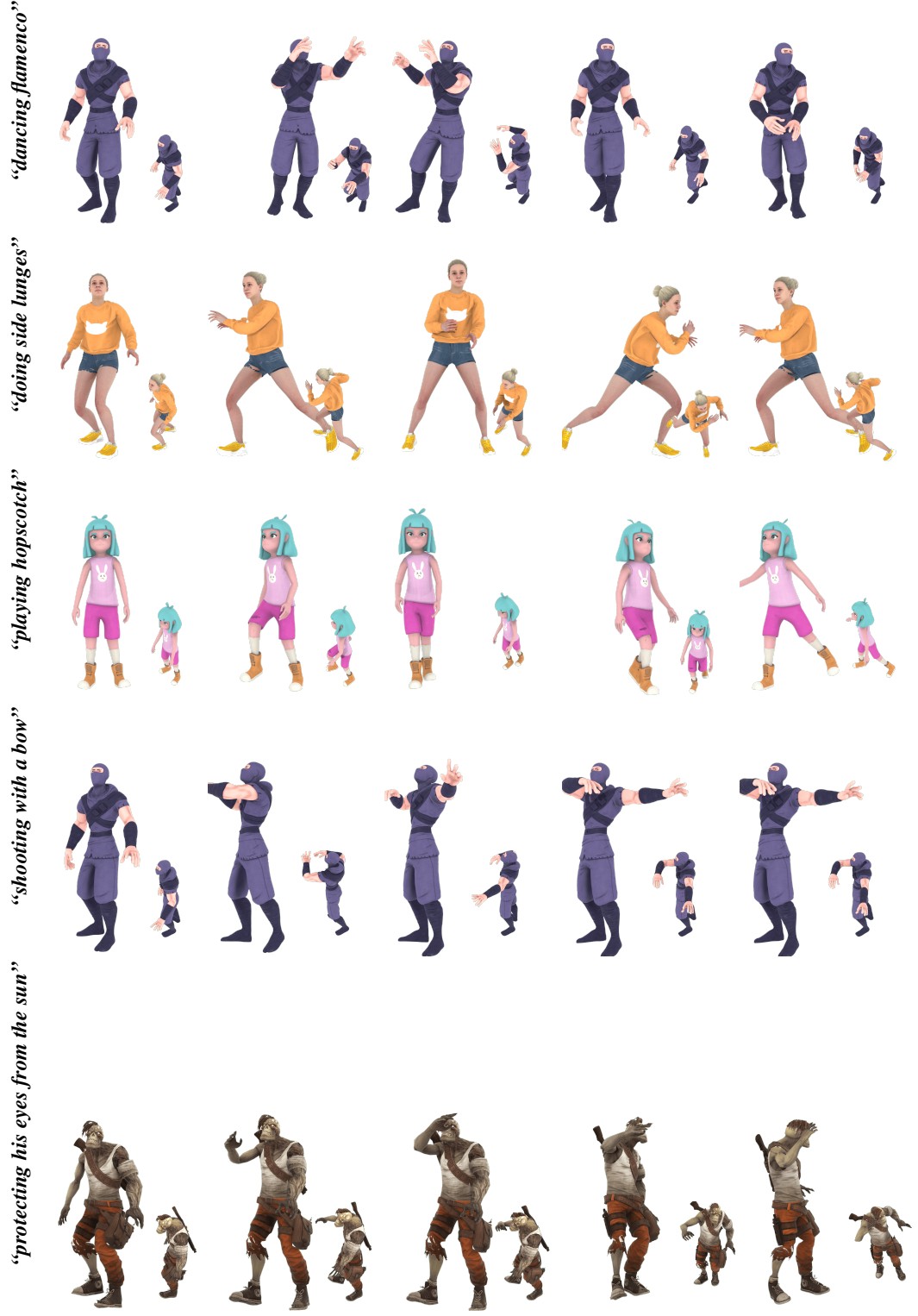

Figure 10: **Additional qualitative results.** Visualization of generated mesh animations with our method. Each row shows: the prompt, the input mesh, and the generated mesh animations. For all generations, we visualize the mesh from the front and side views.

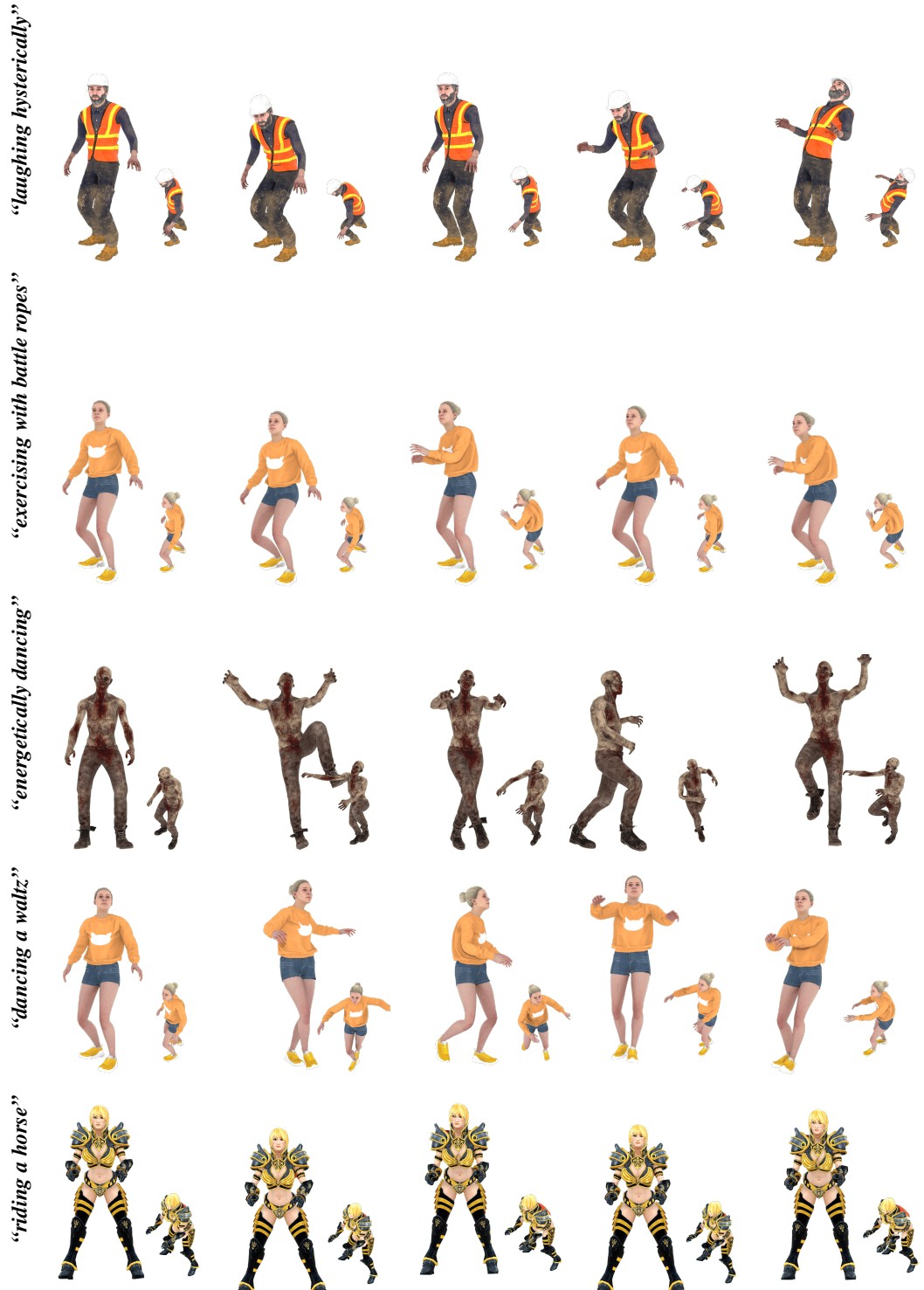

Figure 11: **Additional qualitative results.** Visualization of generated mesh animations with our method. Each row shows: the prompt, the input mesh, and the generated mesh animations. For all generations, we visualize the mesh from the front and side views.

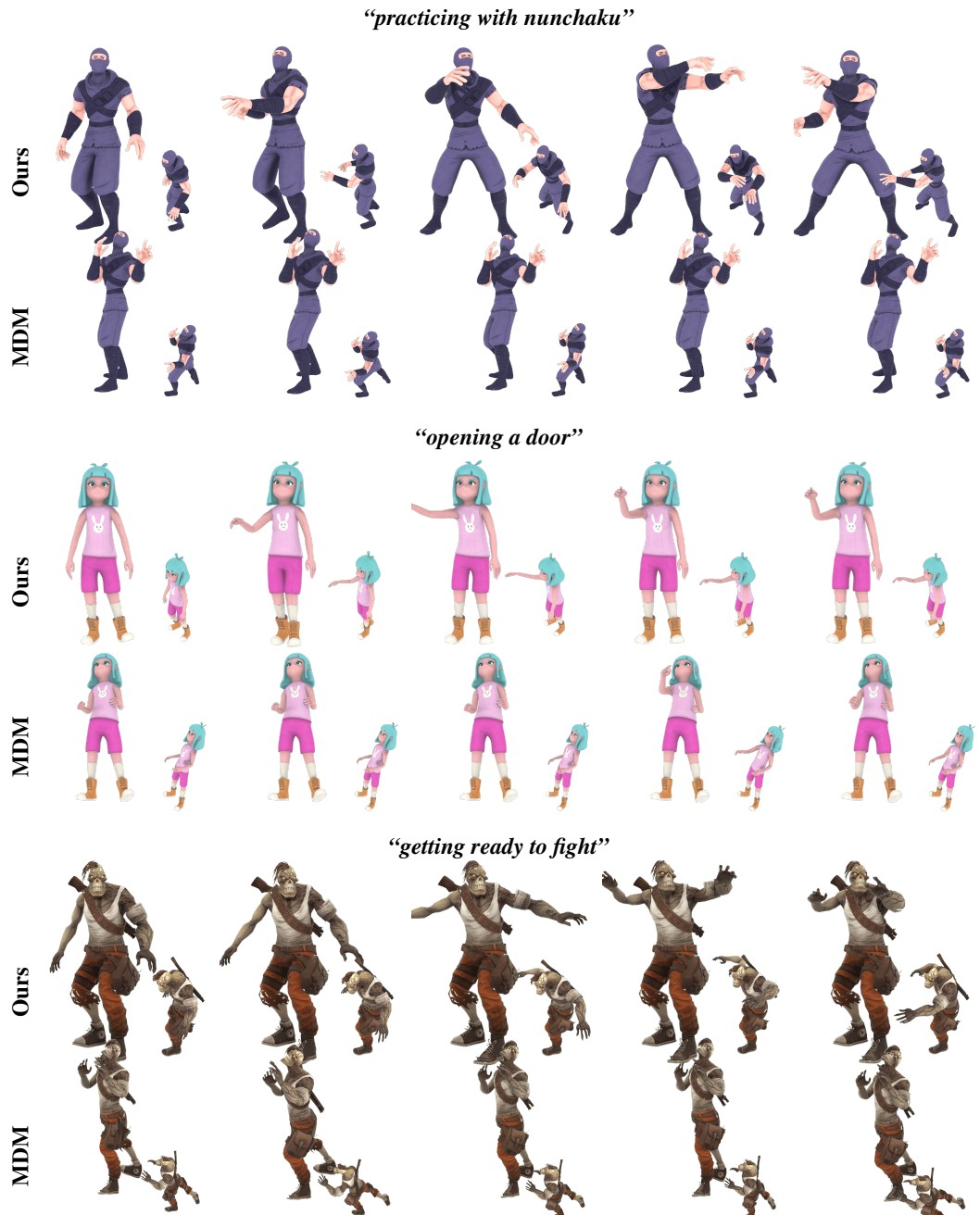

Figure 12: **Additional comparisons with MDM (Tevet et al., 2023).** We show two views (front and side) of the generated motions for multiple frames.

