# OpenReview forum: "Animating the Uncaptured: Humanoid Mesh Animation with Video Diffusion Models"
_ICLR.cc/2026/Conference — ICLR 2026 Poster_

### Official Review · Reviewer_xUv5 · 2025-10-20

**Soundness:** 3
**Presentation:** 3
**Contribution:** 2
**Rating:** 4
**Confidence:** 4

**Summary:**

This paper proposes a new method for animating static 3D humanoid meshes using video diffusion models. Instead of relying on costly motion-capture datasets, proposed approach leverages motion priors learned by large-scale text-to-video diffusion models, which inherently capture diverse human movements. Given a 3D mesh and a text prompt, the model generates a synthetic video of the mesh performing the motion, then reconstructs 3D motion by fitting and optimizing a SMPL body model to track motion cues such as 2D landmarks, silhouettes, and dense DINOv2 features. The optimized SMPL parameters are transferred to the mesh. Experiments on the CAPE dataset show that this method outperforms existing baselines in motion tracking accuracy and smoothness, and a perceptual study confirms that users find the generated animations more realistic and better aligned with textual descriptions.

**Strengths:**

This introduces a simple yet effective framework that leverages the motion priors of existing video diffusion models to animate static 3D humanoid meshes without relying on expensive motion-capture data. Its integration of generative video priors with SMPL-based optimization enables realistic, temporally coherent, and diverse motion synthesis from simple text prompts. This combination, including feature-based optimization, registration, and reparameterization, makes the approach both scalable and generalizable, offering a practical and accessible solution for creating 4D humanoid animations. Also, its performance on pose fitting outperforms the previous works.

**Weaknesses:**

A main concern is that its novelty and contribution are somewhat limited, as the overall concept of using video diffusion models for motion generation has already been explored in prior works such as MotionDreamer [1] and AnyMoLe [2]. Similar to MotionDreamer and AnyMoLe, the proposed approach extracts dense features and generated videos to guide motion reconstruction. Although this paper integrates these components into a clean and unified framework for humanoid, it primarily extends existing ideas rather than introducing a fundamentally new mechanism for motion extraction or representation. Furthermore, while the concepts overlap with [2], this work is not referenced in the paper.


[1] MotionDreamer: Exploring Semantic Video Diffusion Features for Zero‑Shot 3D Mesh Animation. Uzolas et al., 3DV 2025

[2] AnyMoLe: Any Character Motion In‑betweening Leveraging Video Diffusion Models. Yun et al., CVPR 2025

**Questions:**

Why are untextured meshes rendered for video diffusion model inference? It seems that leveraging video diffusion models with textured meshes could yield better performance due to a smaller domain gap with the models’ training distribution. This untextured setting could also effect largely to the pose fitting performance.

---

> ### Author Response · Authors · 2025-11-27
>
> We are thankful for the effort in reviewing our submission.
>
> ### W1. Novelty and related works
> We thank the reviewer for pointing us to AnyMoLe and we have updated the related work section of our document to reflect this relevant work. While we share the motivation of utilizing VDMs generative prior, AnyMoLe targets the distinct task of motion in-betweening and requires expensive per-scene fine-tuning, characters with skeletons and a partial input motion. MotionDremer relies on expensive semantic feature matching from the intermediate activations of the diffusion model which is an expensive process and restricts the duration of the animation and the expressiveness.
>
> ### Q1. Untextured meshes
> We decided to not restrict our method to only textured meshes as they are not always available. However, although all our results are generated from untextured meshes (texture is only added for visualization purposes), our method does still work with textured meshes. We tried using both texted and untextured meshes and in our internal experiments, we didn’t find enough differences that would justify restricting the input to textured meshes.

---

### Official Review · Reviewer_xL5i · 2025-10-30

**Soundness:** 3
**Presentation:** 4
**Contribution:** 4
**Rating:** 10
**Confidence:** 5

**Summary:**

This paper described a method to generate 3D humanoid motion using 2D video generation models. Video models have seen significantly more data compared to motion generation models (that are restricted to motion-capture data typically), and hence seem to be less expressive. The method is well engineered, with individual steps that make perfect sense, and produces convincing results.

**Strengths:**

- excellent results.
- clear and reasonable method.
- some nice steps, such as the combined modalities of the tracking.
- seems to surpass the SOTA even when compared to dedicated motion generation models.

**Weaknesses:**

- Applies to humanoid characters only.
- Depth is not explicitly addressed.

**Questions:**

- How do you not get motions that are too flat in terms of depth?
- Why use VPoser, which is a rather old prior instead of newer ones?
- Why not use the texture of the mesh as well, wouldn't that help the video model to be more expressive?

---

> ### Author Response · Authors · 2025-11-27
>
> We are thankful for the time and effort in reviewing our work.
>
> ### W1. Humanoid characters only
> We decided to target only humanoid characters as there are existing body priors that help during monocular tracking.
>
> ### W2. Depth explicitly not addressed
> By being able to use body priors, the optimization is constrained such that the depth doesn’t need to be explicitly addressed.
>
> ### Q1. Motions with depth
> These body priors, ensures that the generated motions possess realistic depth and are not restricted only to the 2D image plane. Since strictly planar motion would result in anatomically unnatural body poses. Furthermore, we design our prompts (e.g., "dynamic movement") to ensure the VDM generates motions with movement along all directions, providing the tracker with sufficient signal to reconstruct motion along the depth axis.
>
> ### Q2. VPoser and other pose priors
> We experimented with a different and more recent body prior, PoseNDF[1], however, we empirically found VPoser to perform the best. In our internal tests, when using PoseNDF, the elbows tended to bend backwards and go behind the body.
>
> ### Q3. Textured meshes
> We decided to not restrict our method to only textured meshes as they are not always available. However, although all our results are generated from untextured meshes (texture is only added for visualization purposes), our method does still work with textured meshes. We tried using both texted and untextured meshes and in our internal experiments, we didn’t find enough differences that would justify restricting the input to textured meshes.
>
> [1] Tiwari, G., Antic, D., Lenssen, J., Sarafianos, N., Tung, T., & Pons-Moll, G. (2022). Pose-NDF: Modeling Human Pose Manifolds with Neural Distance Fields. In European Conference on Computer Vision (ECCV).

---

### Official Review · Reviewer_FtqR · 2025-11-01

**Soundness:** 3
**Presentation:** 3
**Contribution:** 3
**Rating:** 8
**Confidence:** 4

**Summary:**

This paper presents a novel method to animate a static 3D humanoid mesh from a text prompt, addressing the limitations of costly MoCap datasets by instead leveraging the rich, generalized motion priors from large-scale Video Diffusion Models (VDMs). The pipeline first generates a video by conditioning a VDM on a rendered image of the mesh and the text prompt. Then, it performs a robust motion transfer by first registering a SMPL body model to the input mesh as a "deformation proxy" and re-parameterizing the mesh vertices. A tracker then optimizes the time-varying SMPL parameters to match the generated video, guided by a combination of sparse 2D landmarks, dense silhouettes, and DINOv2 features. To ensure temporal smoothness, these motion parameters are predicted by shallow MLPs, and experiments show this approach outperforms baselines in tracking and is significantly preferred by users over traditional MoCap-based methods for realism and prompt alignment.

**Strengths:**

- The paper's primary strength is its hypothesis that generative video models, trained on massive, diverse, "in-the-wild" video data, contain superior and more generalizable motion priors than the small, clean MoCap datasets currently used by most text-to-motion methods. The strong user study results (Fig. 4) convincingly validate this hypothesis.
- The method for transferring 2D video motion to the 3D mesh is very well-designed. It correctly identifies that regression-based pose estimators (like HMR) would fail on synthetic VDM-generated videos, and thus wisely opts for a more robust optimization-based tracking approach. The use of multiple, complementary tracking cues (sparse landmarks, dense silhouettes, and semantic DINOv2 features) provides strong guidance for the optimization.
- The paper does an excellent job of evaluating its claims. The authors wisely isolate and evaluate their tracking component on a controlled task (recovering GT motion from the CAPE dataset). The results show it is not only more accurate (lower MPJPE/PVE) but significantly smoother (much lower "Accel" error) than strong baselines. The perceptual user study is the main payoff. The fact that users overwhelmingly preferred this method's animations to those from a strong MoCap-based model (MDM) on realism, prompt alignment, and overall quality is a very strong result.
- While the method combines several existing tools (VDMs, SMPL, DINO), it does so in a novel pipeline that solves a practical problem. The "SMPL-as-proxy-rig" approach makes the method applicable to a wide range of static humanoid meshes that lack their own skeletons or rigs, which is a common use case.

**Weaknesses:**

- The method is a "Garbage In, Garbage Out" system that places full trust in the VDM's output. The paper acknowledges that VDMs can produce artifacts or "morphing effects," but does not fully address how the tracker would handle them. If the VDM generates a physically impossible motion, a distorted body part, or a character that morphs into the background, the optimization-based tracker will likely fail or produce an equally nonsensical 3D animation. (I am personally a believer of VDM as world simulators, so hopefully this will be less of an issue over time).
- The tracker works from a single 2D video, which is an inherently ill-posed 2D-to-3D problem. While the SMPL/VPoser prior helps, the method is still susceptible to ambiguities in depth and self-occlusion. The (very specific) prompt engineering in Appendix A (e.g., "Wide angle shot," "Fixed camera," "No zoom in") suggests that the VDM output must be carefully constrained to be "trackable," which limits the range of dynamic camera motions that can be animated.

**Questions:**

N/A

---

> ### Author Response · Authors · 2025-11-27
>
> We are thankful for the effort in reviewing our submission.
>
> ### W1. Dependence on VDM’s output
> A substantial part of our method is designed to stay robust even when the generated videos contain morphing or other artifacts. In such cases, our regularizers push the solution toward physically plausible motions. For instance, if the front of the character morphs into the back, the final animation resolves this by smoothly rotating 180°, rather than collapsing. We have updated the manuscript accordingly and added a visualization of a failure case in Appendix A.1.1. That said, as the reviewer noted, we still rely on the generative quality of VDMs—if the model produces a severely incorrect video, the resulting animation will naturally not look realistic.
>
> ### W2. Ill-posed 2D-to-3D tracking and VDM’s output constraints
> We try to constrain the VDM as much as possible so that it generates videos where the entire body is visible and animated. If only part of the body is shown—for example, just the upper body—the unseen regions cannot be corrected downstream, and the final animation will reflect those missing motions.

---

### Official Review · Reviewer_eAnh · 2025-11-03

**Soundness:** 2
**Presentation:** 2
**Contribution:** 2
**Rating:** 2
**Confidence:** 4

**Summary:**

This paper presents Animating the Uncaptured, a method for animating 3D humanoid meshes from text prompts. Given an input mesh and textual description, the approach first generates motion videos via a text-to-video (T2V) diffusion model, then leverages the SMPL parameterized human model as a deformation proxy to track and reconstruct character motion from the generated video, which is subsequently transferred back to the 3D mesh.
To enhance reconstruction quality, the authors integrate multiple cues of body keypoints, silhouettes, and dense DINOv2 features as optimization constraints. Experiments on the CAPE dataset show that the method outperforms SMPLify-X, WHAM, and Multi-HMR baselines. A user study further indicates that the generated animations achieve higher perceived realism and better text-motion consistency.

**Strengths:**

1. Novelty:
The paper explores a promising direction by introducing a generalized motion prior from large-scale video diffusion models (VDMs) to animate static 3D meshes. The proposed “generation–tracking–deformation” pipeline bridges generative video synthesis and 3D motion reconstruction, leveraging the strong expressive capacity of modern VDMs. This cross-domain integration offers an extensible framework for text-driven 3D animation.

2. Experimental Evidence:
The paper provides abundant qualitative examples, and quantitative results in Table 1 demonstrate clear improvement on CAPE sequences, particularly in the Accel metric, compared with existing registration and reconstruction baselines such as SMPLify-X, WHAM, and Multi-HMR.

**Weaknesses:**

1. Unclear motivation:
While the paper claims to focus on animating humanoid meshes, the methods and experiments seem more centered on registration and tracking from images or videos.
The distinction between animation generation and motion fitting is not clearly articulated, making the actual novelty somewhat ambiguous.

2. Method clarity: Several parts of the method are under-explained.
- line 159 mentions "we use the encoding $Z \in R^32$ of the variational autoencoder VPoser",
but no further elaboration is given.
- lines 256–269 are vague: it is unclear whether $v_i^SMPL$ refers to the corresponding face or vertex on the SMPL template.
The function $\Psi$ is introduced without a precise definition or computational description.
also lacks a clear definition or computational description.
- Equation (3) refers to L1, but the written form corresponds to an L2 prior, indicating inconsistency between text and formulation.

3. Resource requirements:
The paper mentions 1,000 iterations for registration and over 4,000 for video tracking, yet omits device type, runtime, or memory requirements. Without such details, the practical feasibility and scalability of the method remain unclear.

4. Lack of ablation on MLP optimization:
The implicit MLP for temporal modeling appears to be a design simplification rather than a fundamental requirement. Since the MLP is optimized per sequence and does not generalize across meshes, it limits both efficiency and scalability. A shared temporal model (e.g., RNN, Transformer, or motion prior) might offer better generalization and faster inference.

5. Strong Dependence on the Video Diffusion Model:
The presented animations rely heavily on the pretrained Kling AI VDM. The authors neither fine-tune the VDM for animation-related content nor test robustness across different video generators. This raises concerns about reproducibility and generalization to varied video outputs.

**Questions:**

1. Limited quantitative evaluation:
The CAPE dataset provides only narrow evaluation scenarios on untextured meshes.
Why not assess performance on broader human pose and shape benchmarks such as 3DPW, RICH, or EMDB?  Additionally, would textured meshes influence generation quality or tracking efficiency?

2. Tracking accuracy:
Has the tracking accuracy been quantitatively evaluated on editable or rigged humanoid meshes to validate applicability in real animation pipelines?

3. Use of SMPL motion generation:
Since the method already performs mesh-to-SMPL registration, why not leverage existing SMPL-based motion generation techniques to animate the mesh directly, instead of relying on text-to-video tracking? This would seem to be a more straightforward way to drive the mesh using well-established motion priors.

4. Failure cases:
What are the observed failure scenarios?

**Details Of Ethics Concerns:**

The method uses large text-to-video models trained on web-scraped content, which raises data-source and copyright concerns. It could also produce realistic human motion that may be misused for deepfake or impersonation purposes. Ethical safeguards and data transparency are probably insufficiently discussed.

---

> ### Author Response · Authors · 2025-11-27
>
> We are thankful for the effort in reviewing our submission.
>
> ### W1. Motivation and novelty
> Our main novelty is in tackling text-to-motion generation in a zero-shot fashion by using a Video Diffusion Model prior. We do this by tracking the input mesh in the video, so we also evaluate this part of our method comprehensively.
>
> ### W2. Method clarity
> We thank the reviewer for pointing out unclear parts in the text. We have updated the main document to improve the clarity and correctness of the method section (3.1 and 3.3.1).
>
> ### W3. Resource requirements
> Similarly, we have updated section 4.1 (Implementation Details) to include the missing information about runtime and hardware: Our experiments run on 4 CPU cores, 16 GB of RAM, and an NVIDIA RTX 2080 Ti (12 GB VRAM). The full pipeline, which includes visualization and checkpointing during optimization, takes roughly 1.5 hours per sequence.
>
> ### W4. Ablation on MLP optimization
> The motivation for using an MLP rather than directly optimizing pose parameters is to leverage the inductive bias of the network as an implicit temporal prior. We do this in a per-sample optimization to circumvent the limitations of training 4D priors with scarce data. We ablated this design choice by comparing directly optimizing the SMPL parameters or the MPLs and show that the later one achieves better results.
>
> ### W5. Dependence on Video Diffusion Models
> Our method is agnostic to the choice of VDM as it can take the output of different VDMs indistinctly and we don’t make any strong assumption that ties us to a specific model. Since its recent release, we have been able to test the open source VDM Wan2.2 to showcase that our method also works with it. We included the results in appendix A.5.1. Our choice of VDM is due to its superior performance on this task at the time.
>
> ### Q1. Quantitative evaluation and untextured meshes
> As we do not have ground truth tracking available for videos generated by a VDM, we evaluated on the CAPE dataset, which can be rendered to mimic visual characteristics of the VDM output videos, in contrast to other datasets that focus on in-the-wild RGB tracking. Additionally, we now include evaluation on the Mixamo dataset which includes more complex motions.
> We chose not to restrict our task to textured meshes, as they are not always available. However, our method also works with textured meshes, and additional optimization terms could be added to leverage this information. We tried using both texted and untextured meshes and in our internal experiments, we didn’t find enough differences that would justify restricting the input to textured meshes.
>
> ### Q2. Tracking accuracy
> As suggested, we now include a more extensive evaluation (appendix A.7.1) using the Mixamo dataset, which contains rigged humanoid meshes. This evaluation shows that our tracking is still superior to WHAM and SMPLIFY-X when using different identities and more complex motions.
>
> ### Q3. Use of SMPL motion generation
> We chose to not use SMPL-based motion generation methods as we experienced that these works are limited by the data they use for training, which is expensive and hard to capture, whereas video diffusion models' generative prior is richer and more diverse due to the greater amounts of video data available. For example, motions like “digging a hole” or “practicing karate moves” can easily be generated by video diffusion models but not by SMPL-based methods as these motions are not usually captured in the MoCap datasets.
>
> ### Q4. Failure cases
> Thank you for raising this question about the failure scenarios. These can be caused due to prompt adherence, where the VDM generated video does not fit the text prompt, and artifacts in the video, such as limb morphing. However, we incorporate body priors, DINO features, and MPL parameters to tolerate and be robust to these cases. We have included Appendix A.1.1 to showcase these cases.

---

### Meta-Review · Area_Chair_AZRr · 2026-01-10

**Summary:**

The reviews split evenly: two reviewers recommended Reject (eAnh and xUv5), while two recommended Accept (FtqR and xL5i).

Reject camp: eAnh and xUv5 argued the contribution is incremental, overlapping closely with MotionDreamer/AnyMoLe; they flagged missing method details, unreported runtime, narrow evaluation (CAPE only), and limited novelty.

Accept camp: FtqR and xL5i praised the strong user study, solid tracking design, and clear results, seeing the VDM-to-mesh pipeline as a practical advance worth publishing.

**Reviewer Concerns:**

1. Novelty & prior work overlap

– Reviewers worried the idea is too close to MotionDreamer / AnyMoLe; authors clarified task scope (zero-shot, rig-free) and technical differences.

2. Method clarity

– Undefined symbols, L1/L2 mismatch, missing VPoser details; all corrected in revision.

3. Resource & reproducibility

– No runtime/hardware data; authors supplied 1.5 h / RTX 2080Ti and showed the pipeline also runs on Wan2.1.

4. Per-sequence MLP

– Questioned efficiency and generalization; authors added ablation showing smoother motion than direct SMPL optimization and explained data-scarcity constraint.

5. Limited evaluation

– CAPE-only benchmark; authors added Mixamo rigged-mesh results and maintained superiority.

6. Depth & camera constraints

– 2D-to-3D ambiguity and need for careful prompt engineering; authors acknowledged and added failure-case visualizations.

7. Ethics & misuse

– Deep-fake risk inherited from VDM; reviewers satisfied by disclosure and appendix.

**Reviewer Scores:**

N.A.

---

### Decision · Program_Chairs · 2026-01-26

Accept (Poster)